# Heterogeneity of Biofilm Formation Among *Staphylococcus aureus* and Coagulase-Negative *Staphylococcus* Species in Clinically Relevant Intravenous Fat Emulsions

**DOI:** 10.3390/antibiotics14050484

**Published:** 2025-05-09

**Authors:** Gustavo R. Alvira-Arill, Oscar R. Herrera, Jeremy S. Stultz, Brian M. Peters

**Affiliations:** 1Department of Clinical Pharmacy and Outcomes Sciences, College of Pharmacy, Medical University of South Carolina, Charleston, SC 29425, USA; alviraar@musc.edu; 2Department of Clinical Pharmacy and Translational Science, College of Pharmacy, University of Tennessee Health Science Center, Memphis, TN 38163, USA; oherrera@uthsc.edu; 3Department of Pharmacy, Le Bonheur Children’s Hospital, Memphis, TN 38105, USA; 4Department of Clinical Pharmacy and Translational Science, College of Pharmacy, University of Tennessee Health Science Center, Nashville, TN 37211, USA; 5Department of Microbiology, Immunology, and Biochemistry, College of Medicine, University of Tennessee Health Science Center, Memphis, TN 38163, USA

**Keywords:** biofilm, catheter-related infections, fat emulsions, intravenous, parenteral nutrition, *Staphylococcus*

## Abstract

**Background**: Compared to soybean oil intravenous fat emulsion (SO-IFE), use of mixed-oil IFE (MO-IFE) is associated with reduced rates of catheter-related bloodstream infections caused by coagulase-negative *Staphylococcus* species (CoNS) in pediatric patients receiving parenteral nutrition. **Methods**: Using an in vitro biofilm model, this study aimed to assess the impact of IFEs on biofilm formation among *Staphylococcus* species. *S. aureus*, *S. capitis*, *S. epidermidis*, *S. haemolyticus*, *S. hominis*, and *S. lugdunensis* were cultivated as biofilms in media supplemented with SO-IFE, MO-IFE, or fish oil IFE (IFE). Biomass was quantified by the crystal violet method, and follow-up planktonic growth assays assessed antimicrobial effects of IFEs. **Results**: Compared to SO-IFE, MO-IFE and FO-IFE significantly inhibited biofilm formation of *S. aureus* but did not impact planktonic growth. Contrary to clinical data, CoNS biofilm formation was not impacted by any of the IFEs tested. *S. aureus* biofilm inhibition in IFEs was further investigated by comparing differences following growth in SO-IFE supplemented with capric acid, docosahexaenoic acid (DHA), or eicosapenaenoic acid (EPA) to concentrations matching those of MO-IFE. Capric acid supplementation was associated with significant reduction in biofilm formation compared to SO-IFE alone. However, this was attributed to a bactericidal effect based on follow-up planktonic growth assays. **Conclusions**: These results suggest that biofilm formation in *S. aureus* is variably impacted by fatty acid composition in clinically relevant IFEs, with capric acid exhibiting bactericidal activity against tested isolates.

## 1. Introduction

Parenteral nutrition (PN) is indicated for patients with impaired gastrointestinal function and otherwise unable to receive enteral nutrition, usually attributed to critical illness or intestinal failure [1]. While parenteral nutrition administration mitigates malnutrition and is associated with improved outcomes in critically ill patients, it is an independent risk factor for catheter-related bloodstream infections (CR-BSIs) [2,3,4,5]. Pathogens associated with CR-BSIs include *Staphylococcus* species, *Candida* species, and Enterobacterales due to their capacity to form vigorous biofilms on abiotic surfaces like central venous catheters [6,7,8,9,10]. PN composition is tailored to meet each patient’s nutritional needs but generally consists of amino acids, dextrose, and fat emulsion. Prior evidence suggests an independent positive association between fat emulsion inclusion and CR-BSIs, based on promotion of biofilm formation by *Staphylococcus* species and *Candida albicans* [10,11,12,13,14,15].

Prior studies evaluating intravenous fat emulsion (IFE) use and CR-BSI incidence were largely based on soybean oil (SO-IFE; e.g., Intralipid^®^) formulations, but there is increasing utilization of mixed-oil (MO-IFE; e.g., SMOFlipid^®^) and fish oil (FO-IFE; e.g., Omegaven^®^) formulations in the United States, which may impact the etiology associated with PN receipt [16,17]. For example, SO-IFE and FO-IFE were shown to stimulate *C. albicans* biofilm formation in vitro, whereas MO-IFE comparatively inhibited biofilm formation by approximately 50% [18]. Clinically, the receipt of MO-IFE compared to SO- or FO-IFE in pediatric patients receiving PN that developed a fungal CR-BSI was associated with a non-statistically significant reduced proportion of infections caused by *C. albicans*, and a compensatory increase in infections caused by non-albicans *Candida* species [19].

From a single-center analysis, the receipt of MO-IFE compared to SO-IFE in similar patients was associated with a reduced rate of CR-BSIs, but this was largely attributed to reductions in infections caused by coagulase-negative staphylococci (CoNS) [20]. However, it is unclear whether clinically relevant IFEs impact staphylococcal biofilm formation [11,12,21]. The fatty acid composition of clinically relevant IFEs in the US varies as follows: SO-IFE contains high concentrations of ω-6 fatty acids including linoleic acid and linolenic acid, FO-IFE contains high concentrations of ω-3 fatty acids including docosahexaenoic acid (DHA) and eicosapentaenoic acid (EPA), and MO-IFE contains a blend of soybean oil, medium-chain triglycerides (MCT), olive oil, and fish oil [16,17]. Individually, several fatty acids have been found to suppress biofilm formation of *Staphylococcus* species including linoleic acid, linolenic acid, oleic acid, DHA, and EPA, which are variably found in these IFE formulations [22,23,24]. Therefore, this study aimed to assess the impact of clinically relevant IFEs on biofilm formation of *Staphylococcus aureus* and CoNS.

## 2. Results

### 2.1. MO- and FO-IFE Limits Biofilm Formation of S. aureus Isolates Compared to SO-IFE

An established in vitro model system using SO-, MO-, and FO-IFE was employed to examine biofilm growth of *S. aureus* laboratory and clinical isolates. The presence of IFE dose-dependently promoted biofilm formation compared to no-lipid controls irrespective of methicillin or vancomycin sensitivity (Figure 1). A larger panel of representative isolates was further investigated for potential biofilm growth differences in clinically relevant concentrations (5%) of IFEs. Compared to SO-IFE at similar concentrations, growth was reduced with MO-IFE and FO-IFE for most isolates (Figure 2). However, the extent to which isolates formed biofilm was heterogenous.

### 2.2. MO- and FO-IFE Do Not Inhibit Planktonic Growth of S. aureus

To determine whether biofilm repression was attributed to reduced cell growth or specifically biofilm inhibition, *S. aureus* was grown planktonically in 10% SO-IFE, 10% MO-IFE, 5% FO-IFE, or an equivalent concentration of TSB alone. Enumeration of bacterial burden revealed that growth in each IFE was generally similar, suggesting that reduced biofilm formation was not attributed to a bactericidal or bacteriostatic effect on *S. aureus* (Figure 3). Unexpectedly, growth in IFEs did not routinely differ from the no-lipid control, further emphasizing their biofilm-specific impact.

### 2.3. Fatty Acid Composition May Impact Biofilm Formation of S. aureus

IFE formulations differ in their fatty acid composition. For example, MO-IFE differs from SO- and FO-IFE by inclusion of capric and caprylic acid, whereas MO- and FO-IFE both contain DHA and EPA while SO-IFE does not (18). To further investigate whether reduced biofilm formation with MO- and FO-IFE exposure was attributed to differences in concentrations of these fatty acids, select *S. aureus* isolates were grown as biofilms in SO-IFE supplemented with capric acid, DHA, and EPA to concentrations matching those found in MO-IFE. Biofilm formation was significantly reduced with capric acid supplementation compared to SO-IFE alone. However, biofilm growth did not differ with DHA or EPA supplementation (Figure 4). Similar planktonic growth experiments were conducted in SO-IFE supplemented with capric acid, DHA, or EPA to concentrations matching those of MO-IFE to assess whether biofilm repression was simply attributed to reduced cell growth. Supplementation with capric acid was strongly inhibitory (and likely bactericidal) given the lack of observable growth on enumeration plates. DHA and EPA also demonstrated moderate antibacterial effects compared to growth in SO-IFE alone (Figure 5).

### 2.4. IFE Does Not Readily Enhance Biofilm Formation of Most CoNS

As performed with *S. aureus*, an in vitro model was used to assess whether laboratory and clinical isolates of *Staphylococcus capitis*, *Staphylococcus epidermidis*, *Staphylococcus haemolyticus*, *Staphylococcus hominis*, and *Staphylococcus lugdunensis* could form biofilms in SO-, MO-, and FO-IFE. Except for *S. hominis* NRS122 that formed biofilm comparatively well in FO-IFE, the presence of IFE generally did not promote biofilm formation compared to no-lipid controls (Figure 6). Expanding the number of isolates assessed did not reveal IFE-specific biofilm growth trends, save for an additional *S. hominis* isolate (VCU122) that again grew comparatively better in FO-IFE (Figure 7).

## 3. Discussion

Evaluation of biofilm formation in clinically relevant IFEs has largely been limited to SO-IFE as it was the predominant fat source for PN in the US, but newer MO- and FO-IFE have increased in utilization and appear to have an impact on CR-BSI etiology [19,20]. From retrospective cohort studies performed, receipt of MO-IFE appears to be associated with a reduction in CR-BSI rates in pediatric patients receiving PN, which was associated with reduced infections caused by CoNS. However, the impact of newer IFEs on biofilm formation and growth of *Staphylococcus* species has not yet been systematically evaluated. Our experiments illustrated differences in biofilm formation of *S. aureus* based on IFE exposure, where most isolates exhibited improved biofilm growth in SO-IFE as compared to MO- or FO-IFE. However, this finding was not replicated for CoNS isolates tested where mostly moderate differences were observed amongst IFEs. That said, biofilm growth was generally much lower for CoNS isolates as compared to *S. aureus* and may have reduced our ability to identify significant IFE-specific impact.

Biofilm formation is an important virulence determinant for pathogens associated with CR-BSIs as it can enhance resistance to common antimicrobials and catheter lock therapies [8,9,10,25]. Receipt of PN is an independent risk factor of infection as intraluminal contents of macronutrients could provide nutritional substrates for colonized microorganisms [3,4,5]. Prior analyses suggest inclusion of IFE with PN may exacerbate the risk of CR-BSIs based on receipt of SO-IFE [11,12]. IFEs are composed of various fatty acids, and the impact of specific fatty acids on staphylococcal growth and biofilm formation varies [22,23,24]. Fatty acids present in SO-IFE including linoleic, linolenic, oleic, and palmitic acid have previously demonstrated individual reductions in biofilm formation of *S. aureus* with minimal impact on cell growth [23,24]. Similarly, DHA and EPA present in MO- and FO-IFE were also demonstrated to inhibit biofilm formation and cell growth at higher concentrations [22].

Interestingly, medium- and long-chain fatty acids have been described to disrupt quorum sensing in a variety of bacterial species. Both capric and caprylic acid (present in MO-IFE) can interfere with biofilm growth, motility, and exopolysaccharide production in *Pseudomonas aeruginosa* [26]. The LasI and LasR system responsible for production and response to the quorum molecule 3-oxo-C12-homoserine lactone can be bound by capric and caprylic acid, potentially explaining the effect of these fatty acids on pseudomonal virulence. Moreover, myristoleic and palmitoleic acids can reduce biofilm growth of *Acinetobacter baumannii* by similarly disrupting *N*-acyl-homoserine lactone production to impair quorum sensing [27]. Impacts on staphylococcal quorum sensing have also been reported by Lee, et al. [24]. In a survey of 27 fatty acids, petroselinic, vaccenic, and oleic acid were found to significantly impair biofilm growth without affecting planktonic growth. Follow-up studies revealed that petroselinic acid inhibited expression of the master quorum-sensing regulator *agrA*, with concomitant reduction in *RNAIII*, *hla* (encoding for ⍺-hemolysin), and the virulence regulator *saeR*. Therefore, it is likely that single fatty acids, or their combination ratios, present in clinical IFEs may similarly perturb staphylococcal quorum sensing, virulence, or biofilm growth, as reported in this study.

While these fatty acids were previously shown to individually repress biofilm formation, we found that all clinical IFEs generally stimulated *S. aureus* biofilm growth compared to IFE-free controls. However, biofilm formation was comparatively limited in MO-IFE and FO-IFE. MO-IFE is distinct from SO-IFE with inclusion of medium-chain triglycerides, like capric acid, DHA, and EPA, whereas FO-IFE primarily consists of DHA and EPA. Thus, it is reasonable to speculate that differences in staphylococcal biofilm formation observed among IFEs is dependent on their fatty acid composition. In support of this, supplementing capric acid in SO-IFE to mirror MO-IFE reduced *S. aureus* biofilm formation. However, unlike planktonic growth assays with MO-IFE that showed no growth inhibition, capric acid-supplemented SO-IFE completely inhibited *S. aureus* growth. These data suggest a differential impact of fatty acids on biofilm formation by *S. aureus* depending on the total composition of fatty acids present and their corresponding concentrations. It is possible that the antimicrobial effect of such fatty acids (e.g., capric acid) could be enhanced or repressed by the IFE composition. Future studies could address this possibility by quantifying biofilm growth in the presence of capric acid while modulating concentrations or other fatty acids contained in SO-IFE. Additional studies to delineate differential gene expression during growth in IFEs or fatty acids may yield insight into the mechanisms by which DHA, EPA, and capric acid dysregulate biofilm growth. It is also possible that the potency of these antimicrobial compounds is lost during product storage or complexed in a less active state with other moieties present in the IFE mix.

For most CoNS isolates tested, biofilm growth was relatively poor in all IFEs examined. However, a few *S. hominis* isolates, a single *S. epidermidis* isolate, and most *S. lugdunensis* demonstrated modest biofilm formation. Biofilm formation amongst staphylococci isolates is highly heterogenous and differences observed in these experiments may be highly dependent on the underlying isolate utilized [21,22,23,24,28,29,30,31,32,33,34]. For example, most *S. epidermidis* strains formed poor biofilm, except for NRS101, which is the prototypical and well-characterized isolate used to study biofilm growth in these species [23,35,36]. Thus, overall findings may differ considerably if isolates from known CR-BSIs were employed in our model system, as these may have been selected in vivo for enhanced biofilm growth modalities. Furthermore, adjustments to the in vitro model (e.g., alternative medium, extended incubation times, or pre-coating wells with serum or plasma to enhance attachment) may encourage optimal biofilm growth for improved analysis. Regardless, our results suggest that isolate phenotypic heterogeneity is a key contributing factor to disparate staphylococcal biofilm growth in IFEs.

## 4. Materials and Methods

### 4.1. Bacterial Strains and Culture

A total of 45 clinical isolates were used in this study, as listed in Appendix A. This included 5 isolates each of methicillin-susceptible *S. aureus* (MSSA), methicillin-resistant *S. aureus* (MRSA) clade USA100, MRSA clade USA300, vancomycin-resistant *S. aureus* (VRSA), *S. capitis*, *S. epidermidis*, *S. haemolyticus*, *S. hominis*, and *S. lugdunensis* that were obtained from BEI Resources or the American Type Culture Collection. All isolates were maintained as frozen stocks at −80 °C. Prior to use, isolates were sub-cultured onto trypticase soy agar (TSA; BD Difco, Franklin Lakes, NJ, USA) at 37 °C. Single colonies were cultured in 1× trypticase soy broth (TSB; BD Difco) for 24 h at 37 °C in a shaking incubator. Following growth, isolates were washed with phosphate-buffered saline by centrifugation, counted on a hemocytometer, and adjusted to 1 × 10^7^ CFU/mL in 0.5× TSB.

### 4.2. Biofilm Formation

Biofilm formation on polystyrene was performed like previously described methods, with some modifications [37,38]. Briefly, 100 µL of adjusted culture was added to wells of a 96-well cell culture-treated polystyrene microtiter plate (1 × 10^6^ CFU/well). Plates were incubated for 2 h at 37 °C in a humidified chamber to allow for adherence. Nonadherent cells were carefully removed by washing the wells three times with sterile distilled water (dH_2_O). Various concentrations of SO-IFE (Baxter Healthcare Corporation, Deerfield, IL, USA or Sigma-Aldrich, St. Louis, MO, USA); MO-IFE (Fresenius Kabi, Bad Homburg, Germany); or FO-IFE (Fresenius Kabi, Bad Homburg, Germany) at 5%, 2.5%, 1.25%, 0.625%, 0.3125%, and 0.15625% with 0.5× TSB were added to the plate. As a control, biofilms were prepared similarly using 0.5× TSB. Plates were incubated for 24 h at 37 °C in a humidified chamber to allow for biofilm formation.

### 4.3. Biofilm Quantification with Crystal Violet Staining

For crystal violet staining, wells were washed in dH_2_O to remove nonadherent cells, stained with 0.1% crystal violet, and re-washed three times with dH_2_O. After drying, bound crystal violet was resolubilized in 95% ethanol, transferred to a fresh microtiter plate, and the absorbance was read at 570 nm. Wells containing IFE and TSB alone were similarly stained and used to blank subtract from experimental wells. Results are representative of experimental replicates (*N* = 2–3) and reported as raw optical density (OD). Data were presented as the average ± standard deviation (SD) and differences were assessed using two-way ANOVA and Dunnett’s multiple comparison post-test. Significance was set at *p* < 0.05.

### 4.4. Planktonic Growth and Quantitative Growth Assay

Cultures of staphylococci were prepared as described above and adjusted to a final concentration of 1 × 10^5^ cells/mL in 0.5× TSB supplemented with clinically relevant final concentrations of 10% SO-IFE, 10% MO-IFE, or 5% FO-IFE. Cells were grown at 37 °C in a shaking incubator at 200 rpm for 6 h. Cultures were then serially diluted 10-fold and plated onto TSA by the drop plate method [39]. Plates were incubated for 16–18 h at 37 °C in a humidified chamber and colonies were enumerated. Data were presented in CFU/mL as the average ± SD and differences were assessed using two-way ANOVA and Dunnett’s multiple comparison post-test. Significance was set at *p* < 0.05.

### 4.5. Biofilm and Planktonic Growth of S. aureus with Fatty Acid Supplementation

Biofilm and planktonic growth of select *S. aureus* isolates were assessed as described above following growth with 10% SO-IFE supplemented to a final concentration of 55.1 mM capric acid, 6.8 mM DHA, or 7.4 mM EPA. These concentrations were chosen to match those of 10% MO-IFE [18]. Isolates were selected based on differences in biofilm formation associated with IFE exposure identified in prior experiments. Results for biofilm growth were reported as percentage of the relative control (SO-IFE without fatty acid supplementation).

### 4.6. Statistical Analysis

Comparative tests for biofilm and planktonic growth were performed using Graphpad Prism (v10.4.0) as outlined in each figure legend based on groups and distribution of the dataset.

## 5. Conclusions

Pathogens associated with CR-BSIs involving patients receiving PN may vary based on lipid emulsion composition, as previously suggested with clinically relevant IFEs and their impact on biofilm formation of *Candida* species. This study demonstrated heterogeneity in biofilm formation for several *Staphylococcus* species based on growth in SO-, MO-, or FO-IFE. Growth in MO- and FO-IFE limited *S. aureus* biofilm formation compared to SO-IFE, but this pattern varied among CoNS isolates. Supplementation of SO-IFE with capric acid, uniquely present in MO-IFE, strongly repressed biofilm growth, likely due to its bactericidal effect in planktonic culture. Interestingly, this effect was not previously demonstrated on planktonic cultures with MO-IFE alone, which may point toward differential effects of fatty acids on biofilm formation due to combinatorial effects of additional fatty acids or moieties present in IFEs. Future studies are required to further assess mechanisms by which capric acid or complex lipid mixtures drive reduced strain-specific staphylococcal biofilm growth.

## Figures and Tables

**Figure 1 antibiotics-14-00484-f001:**
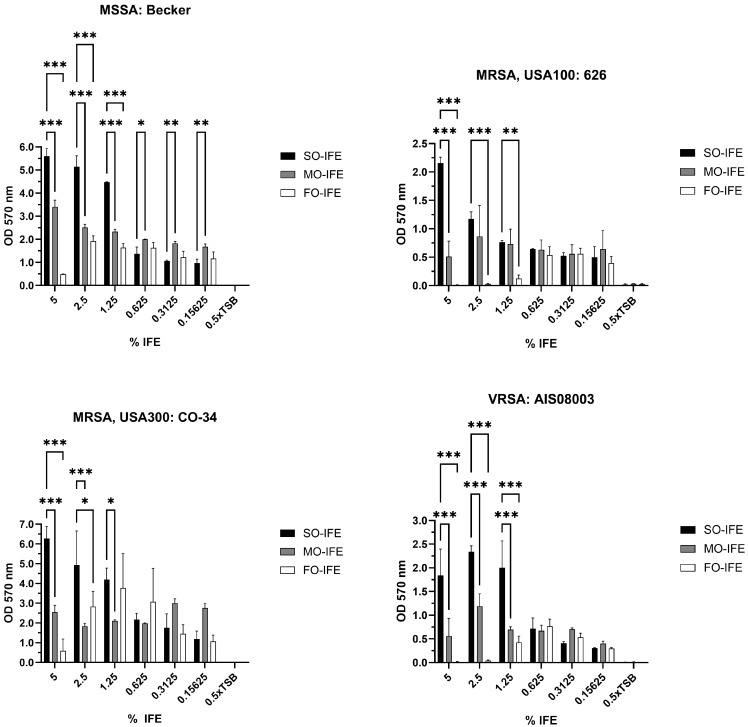
Biofilm formation of select *Staphylococcus aureus* isolates in varying concentrations of SO-IFE, MO-IFE, and FO-IFE. Representative isolates of four categories of *S. aureus*: methicillin-susceptible *S. aureus* (MSSA), methicillin-resistant *S. aureus* (MRSA) clade USA 100, MRSA clade USA 300, and vancomycin-resistant *S. aureus* (VRSA) were cultivated as biofilms for 24 h in 0.5× tryptic soy broth (TSB) supplemented with serial dilutions of soybean oil intravenous fat emulsion (SO-IFE), mixed-oil intravenous fat emulsion (MO-IFE), or fish oil intravenous fat emulsion (FO-IFE). Data expressed as mean ± standard deviation. *N* = 2 replicates. *, *p* < 0.05; **, *p* < 0.01; ***, *p* < 0.001 using two-way ANOVA and Dunnett’s multiple comparison post-test to SO-IFE at each concentration.

**Figure 2 antibiotics-14-00484-f002:**
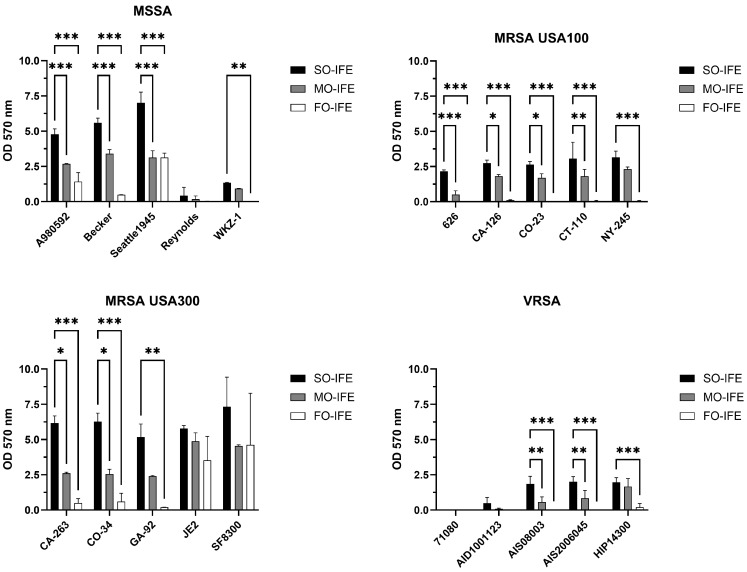
Biofilm formation of an expanded panel of *Staphylococcus aureus* isolates in clinically relevant concentrations of SO-IFE, MO-IFE, and FO-IFE. Laboratory or clinical isolates of *S. aureus* categorized as methicillin-susceptible *S. aureus* (MSSA), methicillin-resistant *S. aureus* (MRSA) clade USA 100, MRSA clade USA 300, and vancomycin-resistant *S. aureus* (VRSA) were cultivated as biofilms for 24 h in 0.5× tryptic soy broth supplemented with 5% soybean oil intravenous fat emulsion (SO-IFE), mixed-oil intravenous fat emulsion (MO-IFE), or fish oil intravenous fat emulsion (FO-IFE). Data expressed as mean ± standard deviation. *N* = 2 replicates. *, *p* < 0.05; **, *p* < 0.01; ***, *p* < 0.001 using two-way ANOVA and Dunnett’s multiple comparison post-test.

**Figure 3 antibiotics-14-00484-f003:**
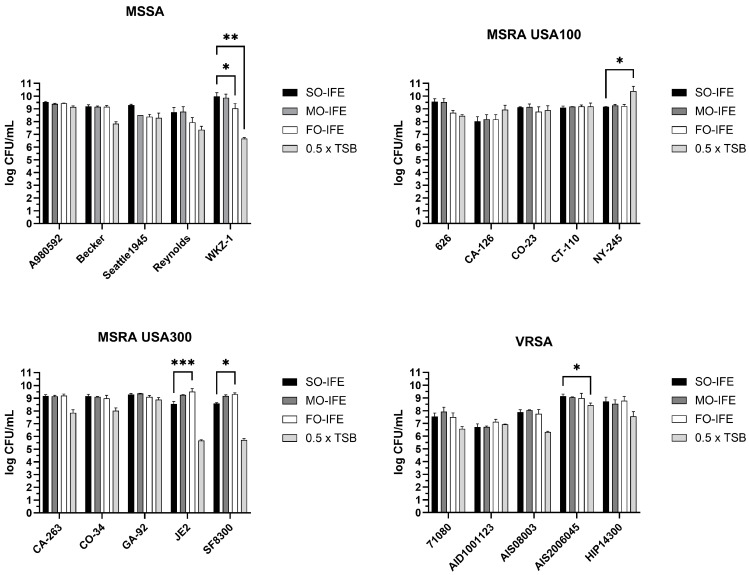
Impact of IFEs on *Staphylococcus aureus* planktonic growth. Planktonic cultures of laboratory or clinical isolates of *S. aureus* categorized as methicillin-susceptible *S. aureus* (MSSA), methicillin-resistant *S. aureus* (MRSA) clade USA 100, MRSA clade USA 300, and vancomycin-resistant *S. aureus* (VRSA) were grown in 0.5× tryptic soy broth (TSB) supplemented with 10% SO-IFE (soybean oil IFE), MO-IFE (mixed-oil IFE), or 5% FO-IFE (fish oil IFE) and subsequently enumerated by drop plate method. Data expressed as mean and standard deviation. *N* = 2 replicates. *, *p* < 0.05; **, *p* < 0.01; ***, *p* < 0.001 using a two-way ANOVA and Dunnett’s multiple comparison post-test.

**Figure 4 antibiotics-14-00484-f004:**
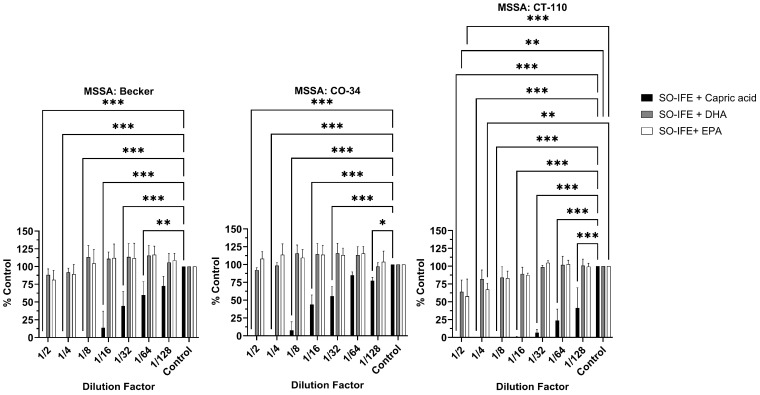
Supplementation of SO-IFE with capric acid significantly reduces biofilm formation in select *Staphylococcus aureus* isolates. Representative isolates of *S. aureus* were cultivated as biofilms for 24 h in 0.5× tryptic soy broth (TSB) and 10% SO-IFE (soybean oil IFE) supplemented with serial dilutions of capric acid, DHA, and EPA at starting concentrations of 58.05 mM, 6.85 mM, and 7.44 mM, respectively. Data expressed as mean ± standard deviation. *N* = 3 replicates. *, *p* < 0.05; **, *p* < 0.01; ***, *p* < 0.001 using two-way ANOVA and Dunnett’s multiple comparison post-test.

**Figure 5 antibiotics-14-00484-f005:**
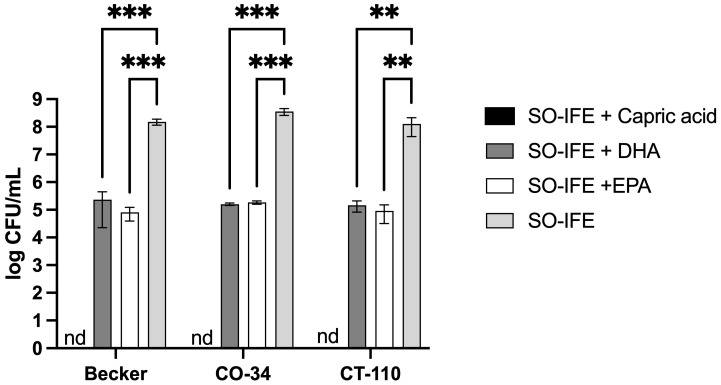
Capric acid supplementation of SO-IFE significantly impacts growth of select *Staphylococcus aureus* isolates. Planktonic cultures of representative isolates of *S. aureus* were grown in 0.5× tryptic soy broth (TSB) and 10% SO-IFE (soybean oil IFE) supplemented with serial dilutions of capric acid, DHA, and EPA at concentrations of 58.05 mM, 6.85 mM, and 7.44 mM, respectively. Data expressed as mean ± standard deviation. *N* = 3 replicates. **, *p* < 0.01; ***, *p* < 0.001 using two-way ANOVA and Dunnett’s multiple comparison post-test. nd, not detected for SO-IFE + Capric acid.

**Figure 6 antibiotics-14-00484-f006:**
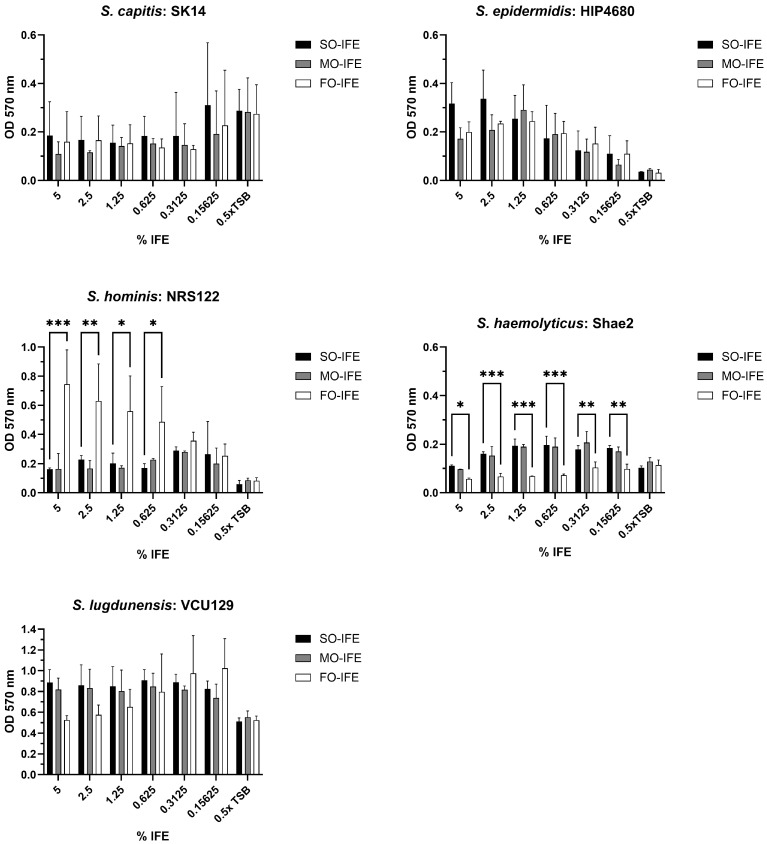
Biofilm formation of select CoNS isolates in varying concentrations of SO-IFE, MO-IFE, and FO-IFE. Representative isolates of five CoNS species: *Staphylococcus capitis*, *Staphylococcus epidermidis*, *Staphylococcus hominis*, *Staphylococcus haemolyticus*, and *Staphylococcus lugdunensis* were cultivated as biofilms for 24 h in 0.5× tryptic soy broth (TSB) supplemented with serial dilutions of SO-IFE (soybean oil IFE, MO-IFE (mixed-oil IFE), or FO-IFE (fish oil IFE). Biomass quantified by crystal violet staining. Data expressed as mean ± standard deviation. *N* = 2 replicates. *, *p* < 0.05; **, *p* < 0.01; ***, *p* < 0.001 using two-way ANOVA and Dunnett’s multiple comparison post-test to SO-IFE at each concentration.

**Figure 7 antibiotics-14-00484-f007:**
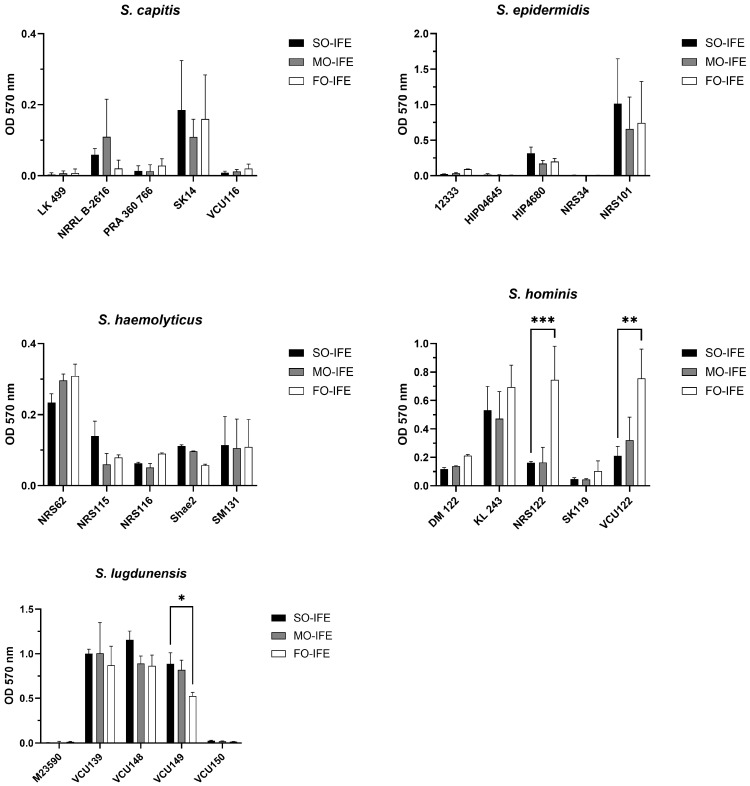
Biofilm formation of CoNS isolates in clinically relevant concentrations of SO-IFE, MO-IFE, and FO-IFE. Laboratory or clinical isolates of CoNS categorized as *Staphylococcus capitis*, *Staphylococcus epidermidis*, *Staphylococcus haemolyticus*, *Staphylococcus hominis*, and *Staphylococcus lugdunensis* were cultivated as biofilms for 24 h in 0.5× tryptic soy broth (TSB) supplemented with 5% SO-IFE (soybean oil IFE), MO-IFE (mixed-oil IFE), or FO-IFE (fish oil IFE) and biomass quantified by crystal violet staining. Data expressed as mean ± standard deviation. *N* = 2 replicates. *, *p* < 0.05; **, *p* < 0.01; ***, *p* < 0.001 using a two-way ANOVA and Dunnett’s multiple comparison post-test.

## Data Availability

All generated data during this study can be made available upon request.

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
