# Peer review of "Heterogeneity of Biofilm Formation Among Staphylococcus aureus and Coagulase-Negative Staphylococcus Species in Clinically Relevant Intravenous Fat Emulsions"

_antibiotics, 2025, doi:10.3390/antibiotics14050484_

Round 1
Reviewer 1 Report
Comments and Suggestions for Authors
Dear Authors,
Staphylococci are microorganisms widespread in the environment, including the hospital environment. Staphylococcus aureus is considered the most pathogenic within this group. However, it is worth noting that on the one hand, coagulase-negative staphylococci, mainly Staphylococcus epidermidis, form part of the human microbiota, but on the other hand they are important etiological agents of infections, especially in immunocompromised patients or in patients with biomaterial implants. Their pathogenicity is due to, among other things, their adhesion abilities, and the production of mucus which can lead to the formation of a biofilm on the surface of tissues but also on biomaterials. Among other things, the biofilm serves as a protective shield against the immune mechanisms of the human body or against compounds with antimicrobial activity, such as antibiotics.
The topic of your paper is interesting and relevant. The choice of strains – staphylococci, including coagulase-negative staphylococci – is appropriate. If you decide to continue your work on this topic, it would be good to extend the research model used to include other methods. The method used has its advantages and disadvantages and it is better not to use it as the only one. It would benefit your research if you also included, for example, microscopic methods such as confocal microscopy. Please explain your reasons for choosing such a research model.
Also, the Authors are asked to address the remaining questions, concerns or suggestions:
- In the Reviewer's opinion, it would be good to present the results as raw figures, e.g. in a supplement. Please consider this suggestion.
- In section 4.2 ‘Biofilm formation’ please complete the literature item you used and at the same time answer the question why the initial incubation was conducted in a moist chamber.
- The ‘Conclusions’ section is missing. Please briefly summarize and interpret the results of the study.
- The species names of bacteria and fungi should be written in italics. The first time a name is mentioned, the full name, e.g. Staphylococcus aureus, Candida albicans, should be used. For subsequent mentions, you can use the abbreviation e.g. S. aureus., C. albicans, etc. Please use the correct way to write microbial names throughout the paper.
Your sincerely
Author Response
Staphylococci are microorganisms widespread in the environment, including the hospital environment. Staphylococcus aureus is considered the most pathogenic within this group. However, it is worth noting that on the one hand, coagulase-negative staphylococci, mainly Staphylococcus epidermidis, form part of the human microbiota, but on the other hand they are important etiological agents of infections, especially in immunocompromised patients or in patients with biomaterial implants. Their pathogenicity is due to, among other things, their adhesion abilities, and the production of mucus which can lead to the formation of a biofilm on the surface of tissues but also on biomaterials. Among other things, the biofilm serves as a protective shield against the immune mechanisms of the human body or against compounds with antimicrobial activity, such as antibiotics.
The topic of your paper is interesting and relevant. The choice of strains – staphylococci, including coagulase-negative staphylococci – is appropriate. If you decide to continue your work on this topic, it would be good to extend the research model used to include other methods. The method used has its advantages and disadvantages and it is better not to use it as the only one. It would benefit your research if you also included, for example, microscopic methods such as confocal microscopy. Please explain your reasons for choosing such a research model.
We thank the reviewer for their important comment. While there are limitations, polystyrene and crystal violet staining are well-established methods for assessing biofilm formation amongst both fungal and bacterial species. The experiments here were adapted from prior published work with Candida species showing a differential impact of IFEs on fungal biofilm growth. Given our previously published clinical report describing IFE-specific increased incidence of catheter-related bloodstream infections for select staphylococcal species, we set forth to recapitulate these findings in an in vitro model system.
Also, the Authors are asked to address the remaining questions, concerns or suggestions:
- In the Reviewer's opinion, it would be good to present the results as raw figures, e.g. in a supplement. Please consider this suggestion.
Based on additional reviewer comments, we have uploaded a new PowerPoint file containing the individual Figures. Raw data will be available upon request or uploaded to a repository as per the Data Availability Statement.
- In section 4.2 ‘Biofilm formation’ please complete the literature item you used and at the same time answer the question why the initial incubation was conducted in a moist chamber.
We have now added additional references that mention the use of a moist chamber for culturing of staphylococcal biofilms. Generally, this procedure is used to prevent evaporation to limit plate “edge effects”.
- The ‘Conclusions’ section is missing. Please briefly summarize and interpret the results of the study.
We thank the reviewer for their helpful comment. We have now added a “Conclusions” section to summarize the work.
- The species names of bacteria and fungi should be written in italics. The first time a name is mentioned, the full name, e.g. Staphylococcus aureus, Candida albicans, should be used. For subsequent mentions, you can use the abbreviation e.g. S. aureus., C. albicans, etc. Please use the correct way to write microbial names throughout the paper.
We thank the reviewer for this important detail and apologize for the oversight. All genus and species names have now been italicized.
Reviewer 2 Report
Comments and Suggestions for Authors
The article “Heterogeneity of biofilm formation among Staphylococcus aureus and coagulase-negative Staphylococcus species in clinically relevant intravenous fat emulsions” presented by Alvira-Arill et al., explores the application of different formulations of fat emulsions applied in parenteral nutrition. It follows a good methodology regarding procedures, in which authors evaluated many bacterial strains in both biofilm formation and planktonic growth. The selection of concentrations for each IFE, as well as the evaluation of capric acid incorporation, are well sustained. However, the presentation of results must improve, and the discussion must be expanded to give better relation between S. aureus and Coagulase negative Staphylococcus species with the different IFEs and fatty acids. Some observations and suggestions are listed below.
Tittle
Scientific names of microorganisms must be in italics.
Abstract
The names of microorganisms must be in italics. Check through the document.
Line 18: In vitro must be in italics. Check through the document.
Results
Line 82: The sentence “Biofilm quantification by crystal violet assay” could be removed. Same in line 92.
Line 104. The sentence “IFEs do not significantly impact S. aureus planktonic growth” could be changed for “Impact of IFEs on S. aureus planktonic growth”. That way, that is the tittle of the figure and latter, the small explanation takes place.
Figure quality must be highly improved. Some figures are hardly visible. If the legend of all the graphs of a figure are the same, it is recommended to only specify it in one figure, that way, authors can optimize size and distribution of the graphs for better viewing.
Discussion
The authors mentioned a possible effect of the IFE composition on the antibacterial activity of capric acid. Please search for a reference that could support this statement.
The discussion presents possible responses about the results obtained, however, it its recommended to expand it. Possible mechanisms regarding molecular biology or quorum sensing could be included, this will increase the impact of the research.
Materials and methods
Line 218: Correct magnitude of cell concentration (number 7 must be superscript). Same in line 222 and 239.
Section 4.2: Please specify whether each concentration of IFE was added to the plate.
Line 241-242: Unit of temperature is not correct.
Erase line 256.
Although the template states that the conclusion is not mandatory, I recommend to include it.
Author Response
Reviewer 2
The article “Heterogeneity of biofilm formation among Staphylococcus aureus and coagulase-negative Staphylococcus species in clinically relevant intravenous fat emulsions” presented by Alvira-Arill et al., explores the application of different formulations of fat emulsions applied in parenteral nutrition. It follows a good methodology regarding procedures, in which authors evaluated many bacterial strains in both biofilm formation and planktonic growth. The selection of concentrations for each IFE, as well as the evaluation of capric acid incorporation, are well sustained. However, the presentation of results must improve, and the discussion must be expanded to give better relation between S. aureus and Coagulase negative Staphylococcus species with the different IFEs and fatty acids. Some observations and suggestions are listed below.
Tittle
Scientific names of microorganisms must be in italics.
We apologize for the formatting oversight and have now italicized all genus and species names.
Abstract
The names of microorganisms must be in italics. Check through the document.
We have edited accordingly as above.
Line 18: In vitro must be in italics. Check through the document.
We have now italicized all instances of “in vitro”.
Results
Line 82: The sentence “Biofilm quantification by crystal violet assay” could be removed. Same in line 92.
Thank you for this comment to improve clarity of the writing.
Line 104. The sentence “IFEs do not significantly impact S. aureus planktonic growth” could be changed for “Impact of IFEs on S. aureus planktonic growth”. That way, that is the tittle of the figure and latter, the small explanation takes place.
Thank you for the helpful comment. This sentence has been edited accordingly.
Figure quality must be highly improved. Some figures are hardly visible. If the legend of all the graphs of a figure are the same, it is recommended to only specify it in one figure, that way, authors can optimize size and distribution of the graphs for better viewing.
We apologize for difficulty in viewing the figures but thank the reviewer for pointing out this important problem. The horizontal and vertical resolution of all figures was set to 600 dpi which meet submission specifications. However, we feel there may have been an issue during rendering of the original manuscript file. The figures have also been submitted as a separate file via PowerPoint with the associated figure legends. The figures will also be maintained in-text for continuity between submission. We have made additional minor formatting edits to better align the graph images for aesthetics and hope they are now sufficiently viewable.
Discussion
The authors mentioned a possible effect of the IFE composition on the antibacterial activity of capric acid. Please search for a reference that could support this statement.
As part of our findings from literature review, IFE composition on antibacterial activity of specific fatty acids has not been thoroughly explored. To date, published work has focused predominantly on individual fatty acids or mixed formulations on biofilm formation of Staphylococcus species, but not on the impact of varying concentrations of multiple fatty acids in media. While we are unable to provide a reference, we feel the statement is still appropriate for the Discussion as a potential explanation for our findings.
The discussion presents possible responses about the results obtained, however, it its recommended to expand it. Possible mechanisms regarding molecular biology or quorum sensing could be included, this will increase the impact of the research.
Additional edits to the Discussion were made to recognize differential expression analyses as an option to assess molecular mechanisms that could contribute to differences in biofilm formation following growth in the different IFE formulations. We have also added additional Discussion points regarding how fatty acids can disrupt quorum sensing activity in a variety of bacterial species, including S. aureus.
Materials and methods
Line 218: Correct magnitude of cell concentration (number 7 must be superscript). Same in line 222 and 239.
Thank you. We have corrected this.
Section 4.2: Please specify whether each concentration of IFE was added to the plate.
Thanks for pointing out this important detail. We have edited accordingly.
Line 241-242: Unit of temperature is not correct.
Thank you. This has now been appropriately corrected.
Erase line 256.
Thank you for noticing this obvious formatting error. This has now been removed.
Although the template states that the conclusion is not mandatory, I recommend to include it.
We agree and have now added a “Conclusions” section to summarize the work.
Reviewer 3 Report
Comments and Suggestions for Authors
This paper studies the effects of three types of IFE on the biofilm formation of Staphylococcus species. Through comparative experiments, it analyzes the effects of these three types of IFE on the biomass and the bacteria number. In addition, the paper examines the influence of the components in MO-IFE on the formation of bacterial biofilms, and concludes that capric acid exhibits bactericidal activity against the tested strains. The thesis has substantial work content. The following are some suggestions:
1、After supplementing capric acid in SO-IFE, bactericidal properties are exhibited, but this property is not present in MO-IFE. The explanation given in the paper is that different components may inhibit or enhance the bactericidal properties of capric acid. In that case, can it also be speculated that capric acid inhibits or enhances the bactericidal properties of other components? Therefore, I think that the bactericidal properties of capric acid cannot be demonstrated solely from these two comparative experiments. Further experiments should be conducted to provide further evidence.
2、The experiments show that MO- and FO-IFE do not inhibit the planktonic growth of S. aureus, but they limit the formation of biofilms compared with SO-IFE. The mechanisms of their effects on bacterial biofilms are not clear in this paper.
3、Figure 4 is relatively small.
Author Response
Reviewer 3
This paper studies the effects of three types of IFE on the biofilm formation of Staphylococcus species. Through comparative experiments, it analyzes the effects of these three types of IFE on the biomass and the bacteria number. In addition, the paper examines the influence of the components in MO-IFE on the formation of bacterial biofilms, and concludes that capric acid exhibits bactericidal activity against the tested strains. The thesis has substantial work content. The following are some suggestions:
- After supplementing capric acid in SO-IFE, bactericidal properties are exhibited, but this property is not present in MO-IFE. The explanation given in the paper is that different components may inhibit or enhance the bactericidal properties of capric acid. In that case, can it also be speculated that capric acid inhibits or enhances the bactericidal properties of other components? Therefore, I think that the bactericidal properties of capric acid cannot be demonstrated solely from these two comparative experiments. Further experiments should be conducted to provide further evidence.
We appreciate the reviewer’s thoughtful comment here. While this hypothesis is viable, we suspect that capric acid is likely the dominant bactericidal agent given its profound impact on planktonic cultures following supplementation of SO-IFE with it compared to DHA or EPA. While we are supplementing SO-IFE with the same amount of capric acid as MO-IFE, we are unsure whether it remains bioactive in the formulation after manufacture. Regardless, future planning involves evaluating the impact of varying concentrations or multiple fatty acids with capric acid across multiple different staphylococcal strains.
- 2. The experiments show that MO- and FO-IFE do not inhibit the planktonic growth of S. aureus, but they limit the formation of biofilms compared with SO-IFE. The mechanisms of their effects on bacterial biofilms are not clear in this paper.
We thank the reviewer for their comment here. The overall purpose of this work was to attempt to recapitulate our clinical findings using a convenient in vitro model system. Moreover, this study was designed to be a comparative analysis between staphylococcal strains and species, which we feel we have performed adequately. While the study lacks definitive mechanistic explanation of the findings, we feel it still contributes to our understanding of heterogeneity of bacterial biofilm growth in clinically relevant lipid emulsions. Current studies are underway to perform transcriptional profiling of staphylococcal biofilms grown in IFEs to identify potential genes of interest that could be genetically targeted for future studies aimed at delineating mechanism. However, these are beyond the scope of this study.
- Figure 4 is relatively small.
We apologize for difficulty in viewing the figures. While they were rendered at an appropriate DPI, there was an obvious issue during rendering of the final merged manuscript file. We have now provided a Powerpoint file with all figures and legends to better assessment. We have made additional edits to improve readability and aesthetics of the figures. We hope they are now satisfactory.
Round 2
Reviewer 1 Report
Comments and Suggestions for Authors
Dear Authors,
Thank you very much for addressing all comments, questions and concerns and making changes to the text. In the reviewer's opinion, the paper can be published in its current form.
Your sincerely
Author Response
Thank you very much for addressing all comments, questions and concerns and making changes to the text. In the reviewer's opinion, the paper can be published in its current form.
We thank the reviewer for their time and additional review of our manuscript.
Reviewer 2 Report
Comments and Suggestions for Authors
Thank you for taking into consideration the observations made. The structure and explanations are better presented. Figures have better viewing, and the expansion of discussion, along with the answers provided by the authors, enriched the paper. Some minor details could be consider.
- Verify if in dH2O, the number should be subscript.
- Line 260: Reference says 340. The temperature unit is not written correctly.
- Figure 6: If it’s possible, reduce the Y-axis scale; this will improve viewing. Same for figure 7.
- Remove references from conclusion.
Author Response
Thank you for taking into consideration the observations made. The structure and explanations are better presented. Figures have better viewing, and the expansion of discussion, along with the answers provided by the authors, enriched the paper. Some minor details could be consider.
We appreciate the reviewers previous suggestions for helping to improve our manuscript. Additionally, we have addressed the following new comments below.
Verify if in dH2O, the number should be subscript.
We have changed all instances of dH2O to dH2O (subscript) as suggested.
Line 260: Reference says 340. The temperature unit is not written correctly.
We apologize for the error here. We have now corrected it to read “[40]”, which is the appropriate reference.
Figure 6: If it’s possible, reduce the Y-axis scale; this will improve viewing. Same for figure 7.
We have improved scaling of the y-axes in Figures 6 and 7 where possible to improve legibility. We hope the reviewer finds these figures now acceptable.
Remove references from conclusion.
We have removed references from the conclusion as suggested.
Reviewer 3 Report
Comments and Suggestions for Authors
This paper studies the effects of three types of IFE on the biofilm formation of Staphylococcus species. Through comparative experiments, it analyzes the effects of these three types of IFE on the biomass and the bacteria number. In addition, the paper examines the influence of the components in MO-IFE on the formation of bacterial biofilms, and concludes that capric acid exhibits bactericidal activity against the tested strains. The thesis has substantial work content.
The author has already answered and revised the questions I raised.
Author Response
This paper studies the effects of three types of IFE on the biofilm formation of Staphylococcus species. Through comparative experiments, it analyzes the effects of these three types of IFE on the biomass and the bacteria number. In addition, the paper examines the influence of the components in MO-IFE on the formation of bacterial biofilms, and concludes that capric acid exhibits bactericidal activity against the tested strains. The thesis has substantial work content.
The author has already answered and revised the questions I raised.
We thank the reviewer for carefully evaluating our revision and are glad to know that their concerns have been adequately addressed.